# A Three-Dimensional Printed Polycaprolactone–Biphasic-Calcium-Phosphate Scaffold Combined with Adipose-Derived Stem Cells Cultured in Xenogeneic Serum-Free Media for the Treatment of Bone Defects

**DOI:** 10.3390/jfb13030093

**Published:** 2022-07-15

**Authors:** Woraporn Supphaprasitt, Lalita Charoenmuang, Nuttawut Thuaksuban, Prawichaya Sangsuwan, Narit Leepong, Danaiya Supakanjanakanti, Surapong Vongvatcharanon, Trin Suwanrat, Woraluk Srimanok

**Affiliations:** 1Department of Oral and Maxillofacial Surgery, Faculty of Dentistry, Prince of Songkla University, Hatyai 90110, Thailand; worapornsup@gmail.com (W.S.); oom.lalita@yahoo.com (L.C.); narit.l@psu.ac.th (N.L.); danaiya.s@psu.ac.th (D.S.); surapong.v@psu.ac.th (S.V.); bomtep.b@gmail.com (T.S.); woraluk.ws@hotmail.com (W.S.); 2Department of Molecular Biotechnology and Bioinformatics, Faculty of Science, Prince of Songkla University, Hatyai 90110, Thailand; sangsuwan.ji@gmail.com

**Keywords:** polycaprolactone, biphasic calcium phosphate, scaffold, adipose, stem cells

## Abstract

The efficacy of a three-dimensional printed polycaprolactone–biphasic-calcium-phosphate scaffold (PCL–BCP TDP scaffold) seeded with adipose-derived stem cells (ADSCs), which were cultured in xenogeneic serum-free media (XSFM) to enhance bone formation, was assessed in vitro and in animal models. The ADSCs were isolated from the buccal fat tissue of six patients using enzymatic digestion and the plastic adherence method. The proliferation and osteogenic differentiation of the cells cultured in XSFM when seeded on the scaffolds were assessed and compared with those of cells cultured in a medium containing fetal bovine serum (FBS). The cell–scaffold constructs were cultured in XSFM and were implanted into calvarial defects in thirty-six Wistar rats to assess new bone regeneration. The proliferation and osteogenic differentiation of the cells in the XSFM medium were notably better than that of the cells in the FBS medium. However, the efficacy of the constructs in enhancing new bone formation in the calvarial defects of rats was not statistically different to that achieved using the scaffolds alone. In conclusion, the PCL–BCP TDP scaffolds were biocompatible and suitable for use as an osteoconductive framework. The XSFM medium could support the proliferation and differentiation of ADSCs in vitro. However, the cell–scaffold constructs had no benefit in the enhancement of new bone formation in animal models.

## 1. Introduction

Over the last decade, the concept of tissue engineering has been applied to bone grafting procedures. Combining bone substitute scaffolds with osteoprogenitor cells or stem cells and some osteo-inductive growth factors is an effective strategy for the enhancement of new bone formation. To our knowledge, the scaffold is still the most important factor. Our techniques of melt stretching and multilayer deposition (MSMD) and melt stretching and compression molding (MSCM) have been successfully used for the fabrication of the polycaprolactone (PCL)–ceramic bone substitute scaffolds [1,2,3,4,5,6]. Both techniques involve filament-based processing that uses PCL–ceramic filaments to fabricate three-dimensional (3D) scaffolds. PCL is a synthetic polyester that was approved by the Food and Drug Administration (FDA) as a medical and drug delivery device. Its biocompatibility has been widely demonstrated in several in vivo and clinical studies [7,8,9,10]. Quantities of the ceramic materials, including biphasic calcium phosphate (BCP) and hydroxyapatite (HA), of up to 30% by weight have been used as fillers in the PCL matrix. Based on our previous studies [1,2,3,4,5,6], scaffolds fabricated using both these techniques are biocompatible and can act as effective osteoconductive frameworks for new bone regeneration in animal models. The PCL–HA scaffold was proved to be a good carrier of bone morphogenetic protein-2 (BMP) [6]. The BMP-soaked scaffolds could be applied to guided bone regeneration models without using conventional xenogeneic bone graft particles. BCP consists of the stable phase of HA and the more soluble phase of β-tricalcium phosphate (β-TCP) [11,12]. The bioactivity of the PCL–BCP scaffolds is achieved because of their ability to release calcium and phosphate ions, which are essential substrates for the bone formation process [1,2]. However, the machines used in both techniques are prototypes with a low productive capacity and possible batch-to-batch variations. For marketing for wider clinical use, the consistency of the scaffold product lines can be improved using 3D printing technology. A major advantage of biomedical 3D printing is that clinical data from computed tomography can be transferred to the computer-printing software, enabling scaffolds to be designed in precise sizes and shapes to fit into the defects. In this study, PCL–BCP filaments were used in a filament-based 3D printing process for the fabrication of PCL–BCP scaffolds.

Several studies [4,5,13,14,15,16,17,18,19] reported promising results in the enhancement of new bone formation when PCL-based scaffolds were combined with osteogenic cells, including primary osteoblasts, mesenchymal stem cells (MSCs) from bone marrow, and dental pulp tissue. Fat tissue is another source of MSCs. Buccal fat pads are a suitable intra-oral source of ADSCs and provide a large amount of fat tissue. The tissue can be easily harvested during routine intra-oral surgical operations of maxillary third molar removal and orthognathic surgeries. The volumes of the fat available for isolating the stem cells are greater, compared with dental pulp and periodontal tissue. Several studies [20,21,22,23,24] demonstrated that adipose-derived stem cells (ADSCs) expressed immunophenotyping markers similar to bone marrow MSCs. Moreover, they could differentiate toward the lineages of different cell types, including osteoprogenitor cells. Broccaioli et al. [25] and Niada et al. [26] demonstrated that there was no difference in immunophenotype, proliferation, and multi-differentiation between ADSCs isolated from buccal fat pads and those from subcutaneous adipose tissue. In a clinical trial, Khojasteh et al. assessed the efficacy of ADSCs from buccal fat pads as an adjunct to autogenous iliac bone block grafting for the treatment of extensive alveolar ridge defects in eight patients and compared the results with those of the control group, which had no cells [27]. The results demonstrated greater new bone formation in the test group when compared with the control group (65.32% and 49.21%, respectively).

Currently, although the use of MSCs in cell-based therapy is accepted for several tissue engineering models and clinical trials, large amounts of the cells are required for each application. Therefore, the small numbers of stem cells isolated from the tissue need to be increased using culture procedures. In general, culturing MSCs in a fetal bovine serum (FBS)-containing medium is a standard protocol for increasing the cells. FBS is suitable for supporting proliferation and differentiation of the cells because it contains many essential nutrients, hormones, and growth factors. However, concerns remain regarding the risks of disease transmission from contamination by animal-originated pathogens and of immunologic reactions from unidentified bovine proteins. The xenogeneic proteins in FBS could be infused into cells because a wash step prior to cell infusion cannot remove their internalized xenogeneic antigens [28]. Moreover, undefined ingredients and batch-to-batch variations can affect the accuracy of research results and therapeutic outcomes [29]. Currently, xenogeneic serum-free media (XSFM) are considered likely to replace the use of FBS for clinical applications. Several studies [30,31,32,33,34,35,36] demonstrated that XSFM have the capacity to maintain the morphologies, the expression of phenotypic surface markers, and multipotent differentiation of MSCs. Therefore, culturing stem cells in XSFM is expected to become a standard protocol in the large-scale expansion of this technique for clinical use. In our study, the growth and osteogenic differentiation of ADSCs isolated from buccal fat pads when seeded on the PCL–BCP 3D printed (PCL–BCP TDP) scaffolds and cultured in XSFM were evaluated in vitro. In addition, the efficacy of the cell–scaffold constructs for the enhancement of new bone regeneration was assessed in rat models.

## 2. Materials and Methods

PCL pellets (Purasorb PC12, Mn 79,760, Viscometry 1.0–1.3 dL/g) were purchased from Corbion, the Netherlands. BCP particles (HA/β-TCP = 30/70%, particle size < 75 µm) were supplied by the National Metal and Materials Technology Center (MTEC), Pathumthani, Thailand.

### 2.1. Scaffold Fabrication

The PCL pellets and BCP particles were mixed in a ratio of PCL: BCP at 70:30 by weight in the chamber of a melting–extruding machine [1]. A homogenous PCL–BCP blend was obtained by heating and stirring at 120 °C, and then the blend was extruded through the nozzle tip of the machine to form the filament. The architectures of the PCL–30%BCP TDP scaffold were designed in a grid pattern with 500 µm^2^ between the filament rows and at 0°, 45°, and 90° to each lay-down layer using 3D Slicer Software (ideaMaker version 4.1.0.4990, Raise-3D Technologies Inc., Irvine, CA, USA) (Figure 1). The printing parameters included layer height 0.25 mm, infill density 40%, and infill angle 0°/45°/90°. To fabricate the scaffold, the filament was loaded into the 3D printer (RAISE3D-E2, Raise-3D Technologies Inc., Irvine, CA, USA) and extruded through the 0.4 mm nozzle tip of the machine using an extruder temperature at 180 °C and printing speed at 30 mm/s. The scaffolds were placed in sterilization pouches and sterilized using ethylene oxide gas (ethylene oxide 100%, 37 °C, humidity 76%, 2 h) 2 weeks prior to the subsequent experiments.

### 2.2. Scaffold Morphologies and Structural Analysis

The microscopic architectures of the scaffold were evaluated using a stereomicroscope (Nikon, Tokyo, Japan) and a scanning electron microscope (SEM, JOEL Ltd., Tokyo, Japan). The scaffolds were stained with Alizarin Red S (AR, Sigma-Aldrich Inc., St. Louis, MO, USA) for detecting the areas where the BCP particles had deposited using a fluorescent microscope (ZEISS Axio Observer 7, Carl Zeiss, Oberkochen, Germany). The dispersion of the BCP particles in the PCL matrix of the scaffold filaments was assessed using field emission SEM (FE-SEM, Apreo, Thermo Fisher Scientific, Waltham, MA, USA) and energy-dispersive X-ray spectroscopy (EDX, Apero, Thermo Fisher Scientific, Waltham, MA, USA). Functional groups and the chemical interaction between the BCP particles and the PCL matrix were analyzed using Fourier transform infrared spectroscopy (FTIR, Bruker Vertex70, Billerica, MA, USA).

### 2.3. Mechanical Testing

The 10 × 10 × 5 mm^3^ scaffold specimens were immersed in phosphate buffer saline (PBS) and incubated at 37 °C for 24 h before the experiments. Compression tests were applied to the superior and lateral aspects of the specimens in the wet stage using a universal testing machine (Lloyd Instruments Ltd., West Sussex, UK) (n = 5/aspect) (Figure 2). For the superior aspect, each specimen was placed on the flat testing platform against the compressing probe (15.77 mm diameter; 5 kN load cell). Then, compression force was applied to its superior aspect from 0 to 300 N at a crosshead speed of 10 mm/min. For the lateral aspect, the lateral aspect of each specimen was placed against the compressing probe (15.77 mm diameter; 250 N load cell) and secured using a vice grip. Compression force was applied to its lateral aspect at a crosshead speed of 10 mm/min until the strain level reached 30%. The compressive strength of the scaffolds was analyzed using analysis software (NEXYGEN, Lloyd Instruments Ltd., Hampshire, UK).

### 2.4. Subject Enrollment

The experimental protocol was approved by the human research ethics committee, Faculty of Dentistry, Prince of Songkla University (EC6012-37-P-LR). The six volunteers were patients undergoing either surgical removal of impacted maxillary third molars or orthognathic surgeries for correction of skeletal discrepancies, in the Oral & Maxillofacial Surgery clinic, Dental Hospital, Faculty of Dentistry, Prince of Songkla University. The inclusion criteria for the surgical removal of the third molars were healthy men or women from 20 to 40 years old, weighing more than 50 kg. The inclusion criteria for the orthognathic surgeries were healthy men or women (ASA class I) from 20 to 40 years old, weighing more than 50 kg, and with a hematocrit level of at least 35%. The exclusion criteria included patients with a history of radiation of the head and neck region, diabetes, uncontrolled metabolic diseases, compromised immune system, blood-transmitted diseases, infection of surgical sites, postliposuction of buccal regions and pregnancy. All patients provided informed consent prior to the experiments.

### 2.5. Isolating ADSCs from Fat Tissue

For the removal of the third molars, triangular flap incisions were created extending to the buccal vestibule of maxillary first molars. After the impacted teeth were removed, blunt dissection was made through the buccinator muscle, and some parts of the buccal fat tissue were excised. For the orthognathic surgeries, the buccal fat tissue often leaked during Lefort I osteotomy of the maxilla or bilateral sagittal split ramus osteotomy (BSSRO) of the mandible without blunt dissection or force traction. Some parts of the tissue in the operation fields were excised. The fat tissue harvested from each patient was divided equally into 2 groups

XSF group: The fat issue was stored in XSFM (MesenCult™-XF, STEMCELL Technologies Inc, Vancouver, BC, Canada).FBS group: The tissue was stored in Dulbecco’s Modified Eagle Medium (DMEM, Gibco, Thermo Fisher Scientific, Waltham, MA, USA) supplemented with 10% FBS (Gibco, Thermo Fisher Scientific, Waltham, MA, USA).

The stem-cell-isolation procedure for both groups was performed within 2 h. In brief, the fat tissue was washed several times with sterile PBS to remove contaminating debris and red blood cells. The fat tissue was then minced into small pieces and enzymatically digested using 3 mg/mL type I collagenase (Gibco, USA) in PBS at 37 °C with gentle agitation for 60 min. The cell pellet was obtained by centrifugation at 1200 g for 10 min and then resuspended in the medium of each group. The solution was filtered through a 100 µm cell strainer (Corning, Merck KGaA, Darmstadt, Germany) and then plated into 6-well plates (Corning, Merck KGaA, Darmstadt, Germany). The plates were incubated in a humidified atmosphere with 5% CO_2_ at 37 °C until the adherent cells reached 70–80% confluence, and then subculture was performed. The cells from passages 2 to 5 were used for the experiments (Figure 3).

### 2.6. Characterizing ADSCs

#### Flow Cytometry Analysis

The MSC immunophenotypes of the cells were defined following the protocol of the International Society for Cellular Therapy (ISCT) [37]. The analysis was performed using a fluorochrome-conjugated monoclonal antibody cocktail in the MSC Phenotyping Kit, human (Miltenyi Biotec, Bergisch Gladbach, Germany). In brief, 5 × 10^5^ cells in passages 2–3 were incubated in antibodies against the surface antigens CD73, CD90 and CD105 as the positive markers and CD14, CD19, CD34 and CD45 as the negative markers. In addition, antibodies against CD 271 and CD 146 were included in the sequences. At least 10,000 events were acquired for each sample using a fluorescence-activated cell sorting instrument (FACSCalibur, BD Biosciences, Franklin Lakes, NJ, USA) and the data were analyzed using CELLQUEST software (version 4, BD Biosciences, Franklin Lakes, NJ, USA).

### 2.7. In Vitro Proliferation and Osteogenic Differentiation of ADSCs Seeded on the PCL–BCP TDP Scaffolds

#### 2.7.1. Assessment of Cell Proliferation

Prior to cell seeding, the scaffolds were placed in 48-well plates (Corning, Merck KGaA, Darmstadt, Germany) and immersed in the medium of each group for 24 h. The cell–scaffold constructs were obtained by seeding 2 × 10^4^ of the ADSCs of the XSF and FBS groups onto each scaffold (diameter 11 mm and height 2 mm) using the static seeding method. The constructs were left for 24 h in a humidified atmosphere with 5% CO_2_ at 37 °C to allow cell attachment. Next, they were moved to new wells and 200 µL of each group’s medium was added. The constructs were cultured in a humidified atmosphere with 5% CO_2_ at 37 °C, and the media were changed every 3 days until the time of the test. At days 3, 7, 14 and 21 after seeding, the quantity of the viable cells in the constructs was measured using PrestoBlue reagent (Thermo Fisher Scientific Inc., Waltham, MA, USA) (n = 6/group/time point). The medium in each well was removed and the constructs were washed using phosphate buffer saline (PBS), then replaced with 180 µL of fresh media without FBS. Twenty microliters of PrestoBlue reagent was added to each well and the constructs were incubated in 5% CO_2_ at 37 °C for 10 min while protected from direct light. After incubation, 100 µL of the medium in each well was transferred to a 96-well plate in duplicate and the absorbance of each well was read at 570 nm using a microplate reader (Thermo Fisher Scientific, Vantaa, Finland). The levels of OD were compared with a standard curve to infer the quantities of the cells. At each time point after the measurement, the constructs were refreshed with 200 µL of each group’s medium and the culture was continued until the next time point.

#### 2.7.2. Assessment of Cell Differentiation

At 21, 14 and 7 days prior to the experiment, the cell–scaffold constructs were obtained by seeding 1 × 10^6^ of the ADSCs of the XSF and FBS groups onto each scaffold (diameter 11 mm and height 2 mm) using the static seeding method as previously described (n = 6/group/time point). They were left for 24 h in a humidified atmosphere with 5% CO_2_ at 37 °C to allow cell attachment. Next, the constructs were moved to the new wells and 200 µL of osteogenic differentiation (OS) media was added as follows.

XSF–OS group: cultured in xenogeneic serum-free OS medium (MesenCult™ Osteogenic Differentiation Human, STEMCELL Technologies Inc., Vancouver, BC, Canada).FBS–OS group: cultured in DMEM OS medium (DMEM supplemented with 10%FBS, 10 mM β-glycerophosphate (Sigma-Aldrich Inc., St. Louis, MO, USA), 10^−7^ M dexamethasone (Sigma, city, state, USA) and 50 μM ascorbic acid-2 phosphate (Sigma-Aldrich Inc., St. Louis, MO, USA).Control group: The osteoblasts (MC3T3-E1 cell line, subclone 4, ATCC, Manassas, VA, USA) were seeded onto the scaffolds using the same ADSC protocol and cultured in the FBS–OS medium.

The constructs were cultured in a humidified atmosphere with 5% CO_2_ at 37 °C, and the media were changed every 2 days until the time of the test. On the day of the experiment, a quantitative reverse transcription polymerase chain reaction (RT-qPCR) was performed to assess the osteogenic differentiation genes. In brief, the total RNA was extracted and purified using a PureLink™ RNA Mini Kit (Invitrogen, Thermo Fisher Scientific, Waltham, MA, USA) and then reverse-transcribed using a SuperScript III First-Strand Synthesis System (Invitrogen, Thermo Fisher Scientific, Waltham, MA, USA). RT-qPCR was performed using a SensiFAST™ SYBR^®^ No-ROX Kit (Meridian Bioscience, London, United Kingdom), a LightCycler system (Roche Diagnostics, Mannheim, Germany), and specific primers for the osteoblast-related genes as listed in Table 1. The RT-PCR condition was performed via 45 cycles of denaturation, annealing and elongation. The expression of the genes at each time point was analyzed using the 2^−∆∆CT^ method and normalized with the expression of glyceraldehyde 3-phosphate dehydrogenase (GAPDH) housekeeping gene.

#### 2.7.3. SEM

At days 3, 14 and 21 after seeding, the characteristics of the cells in the constructs of the XSF–OS and FBS–OS groups were descriptively assessed. The constructs were removed from the culture plates, rinsed with PBS and then fixed in 2.5% glutaraldehyde (Sigma-Aldrich Inc., St. Louis, MO, USA) in PBS for 2 h. The specimens were dehydrated in the 50–100% ethanol series and coated with gold–palladium. The constructs were observed using SEM.

### 2.8. Assessment of Efficacy of the Cell–Scaffold Constructs for Repairing Calvarial Defects in Rat Models

#### 2.8.1. Preparing the Cell–Scaffold Constructs

The cell–scaffold constructs of the XSF–OS group were obtained by seeding 2 × 10^6^ of the pooled ADSCs onto each scaffold (diameter 8 mm and height 1 mm) using the protocol previously described. The constructs were left to cultivate in the XSF–OS medium for 7 days prior to surgical implantation. 

#### 2.8.2. The Animals

Forty-eight adult male Wistar rats, weighing 300–400 g (Nomura Siam International, Bangkok, Thailand), were used for the experiment. This was approved by the animal experiment ethics committee of the Prince of Songkla University (MHESI 68014/860). The animals were anesthetized using ketamine (60 mg/kg) and xylazine (10 mg/kg), administered intraperitoneally. Next, a bi-cortical calvarial defect (8 mm in diameter) was created in the mid-sagittal plane of each animal. In Group A, the cell–scaffold construct was placed into the defect and covered with resorbable membrane (OssGuide, noncrosslinked collagen membrane, SK bioland Co., Ltd., Seoul, Korea). In Group B, the scaffold without the cells was placed into the defect and covered with the membrane. In Group C, the defects were filled with autogenous calvarial bone chips and covered with the membrane. In Group D, the defect was left empty and covered with the membrane (Figure 4). The wounds were closed with 4/0 absorbable sutures (Vicryl, Ethicon LLC, London, UK). All animals received postoperative antibiotic prophylaxis with subcutaneous cephalexin, 10 mg/kg, and postoperative analgesic with subcutaneous buprenorphine, 0.1 mg/kg once daily for 3 days. At the time points of 2, 4 and 8 weeks after the operation, the animals were sacrificed using intraperitoneal administration of 120 mg/kg of overdose pentobarbital sodium. The calvarial specimens, including the 3 mm margin of normal bone surrounding the areas of the bone defects, were collected and then fixed in 10% formalin for microcomputed tomography (µ-CT) and histological assessment (n = 4/group/time point).

#### 2.8.3. µ-CT Analysis

The specimens were scanned using a µ-CT machine (µ-CT 35, SCANCO Medical AG, Wangen-Brüttisellen, Switzerland) in a direction parallel to the coronal aspect of the calvariums, at settings of 55 kVp, 72 mA and 4 W. The gray-scale threshold values were adjusted to discriminate between new bone and the ceramic particles in the scaffolds. New bone formation within each implant site was measured as the new bone volume fraction (VF) using analysis software (µ-CT 35 Version 4.1, SCANCO Medical AG, Wangen-Brüttisellen, Switzerland) with the following formula:New bone VF = [New bone volume÷Total defect volume] × 100

#### 2.8.4. Histologic Processing and Histological Assessment

The specimens were decalcified in 14% ethylenediaminetetraacetic acid (EDTA) and embedded in paraffin. Serial 5 µm thick sections were cut at positions 500 µm from the midline of each specimen. The sections were stained with hematoxylin and eosin (H&E) (2 sections/specimen). The stained sections were scanned using a slide scanner (Aperio, Leica Biosystems, Deer Park, IL, USA) to create image files. The microscopic features were assessed descriptively.

### 2.9. Statistical Analysis

The chemical and mechanical properties of the scaffolds, characteristics of the cell–scaffold constructs and histological features were descriptively evaluated. The measurable parameters, which included the number of viable cells in the constructs, the levels of the osteogenic genes and the new bone VF, were analyzed using statistical analysis software (SPSS, version 14, IBM Corporation, Armonk, NY, USA). One-way analysis of variance (ANOVA) followed by a Tukey HSD test were applied to assess the differences between the groups and time points. The level of statistical significance was set at *p* < 0.05.

## 3. Results

### 3.1. Scaffold Morphologies

The architectures of the scaffolds are demonstrated in Figure 5. The SEM images demonstrated irregular surfaces of the scaffolds that had a few of the BCP particles depositing (Figure 6A–C). The AR staining indicated that the BCP particles were distributed throughout the surfaces of the scaffolds (Figure 6D).

### 3.2. Structural Analysis of the Scaffolds

The structural bands of the PCL–BCP composite were demonstrated via the FTIR spectra (Figure 7). The spectrum of the composite was slightly different from the PCL and BCP spectra, which indicated that there was no chemical bond between the materials. The FE-SEM demonstrated immiscible dispersion of the BCP particles throughout the PCL matrix and several voids inside the filaments (Figure 8A,B). The EDX analysis indicated calcium–phosphate crystals on the scaffold surfaces (Figure 8C).

### 3.3. Mechanical Properties

The mechanical properties of the scaffolds are shown in Table 2. The scaffolds could successfully withstand compression forces from the superior and lateral directions. They recovered to their initial height without distortion after the forces had been applied.

### 3.4. Demographic Data

Two males and four females, at an average age of 25.17 ± 5.64 years old, were enrolled in the study. There were two cases involving third molar removal and four cases involving orthognathic surgeries (Table 3). All patients tolerated the operation well without postoperative complications. The average fat volume was 4.17 ± 0.98 milliliters.

### 3.5. Cell Morphologies

After plating the cell suspension of the XSF and FBS groups, adherent cells could be detected within 7 days, and they gradually proliferated over time. The cells of both groups had spindle-shaped fibroblast-like morphology (Figure 9). By observation, the cells in the XSF group were more spindle-shaped and grew at higher density compared with those in the FBS group.

### 3.6. Flow Cytometry Analysis

Expression of the MSC immunophenotypes of the ADSCs is shown in Table 4 and Figure 10. The percentages of the CD 73, 90 and the hematopoietic markers of both groups were not statistically different. However, the percentage of the CD 105 in the FBS group was significantly higher than that of the XFS group. The cells in both groups expressed CD73 at the highest levels, followed by CD90 and CD105, and they expressed the hematopoietic markers at less than 1%. No statistical difference was detected between the two groups for the expression of CD271 and CD146.

### 3.7. Cell Proliferation

The proliferation of the ADSCs in the cell–scaffold constructs in the XSF and FBS groups is shown in Figure 11. The cells in both groups proliferated over time until reaching their maximum growth on day 14, and then the growth of the cells decreased on day 21. The overall growth of the cells in the XSF group was notably higher than it was in the FBS group and this significant difference was detected on day 14 (*p* < 0.05).

### 3.8. Cell Differentiation

Expression profiles of the osteoblast-related genes are shown in Figure 12. On day 7, the osteogenic genes in the XSF–OS group, with the exception of Col-1, upregulated to a higher level than those in the FBS–OS and control groups. The genes in both groups downregulated significantly on day 14.

### 3.9. Morphologies of the Cell–Scaffold Constructs

The SEM pictures demonstrate the behaviors of the ADSCs in the cell–scaffold constructs in the XSF–OS and FBS–OS groups (Figure 13). After seeding, the cells attached and grew well throughout the surfaces of the scaffolds. The cells continued to form multilayer cell sheets covering the entire surfaces over 21 days. By observation, there was no difference in the behaviors of the cells in both groups.

### 3.10. Experiment In Vivo

All animals tolerated the operation well and were healthy during the observation period. The surgical wounds healed without complications or signs of foreign body reactions.

#### 3.10.1. µ-CT Analysis

The new bone volume fractions are shown in Figure 14. Over the observation period, the new bone volumes in Groups A–C were notably greater than those in Group D (*p* > 0.05). Significant differences were detected between Groups C and D (*p* < 0.05). At week 4 and 8, the new bone volumes in Group C were notably greater than those in Groups A and B. Significant differences were detected between Groups A and C (*p* < 0.05). Interestingly, the new bone formation in Group A was less than that in Group B at all time points (*p* > 0.05). The 3D-constructed images demonstrated that from week 4, the newly formed bone in Groups A–C occurred in the middle portions of the defects, whereas in Group D, it was located at the peripheries. At week 8, the newly formed bone in Groups A–C filled most areas of the roofs of the defects (Figure 15).

#### 3.10.2. Histological Assessment

Histological features of the implanted areas are shown in Figure 16, Figure 17 and Figure 18. During histological preparation, the scaffolds in both groups totally dissolved as a result of the histological processes; therefore, the scaffolds were observed as empty spaces within the defects. At week 2 (Figure 16), the scaffolds of Groups A and B were surrounded by dense fibrous tissue. Chronic inflammatory cells were generally found infiltrating around the areas of the collagen membranes rather than the scaffold areas. New bone regeneration was detected extending from the periphery host bone of all groups. At week 4 (Figure 17), the infiltration of the inflammatory cells clearly reduced. Areas of new bone formation in the defects of Groups A–C increased notably compared to week 2. In Groups A and B, the newly formed bone regenerated along the roofs of the defects, whereas in Group C, it was found surrounding the bone graft fragments and in the middle portions of the defects. By observation, the collagen formation within the defects in Group C was denser than that in Groups A and B, which had more adipose tissue in their connective tissue stroma. In Group D, newly formed bone continued to regenerate from the peripheries of the defects. There was less collagen formation in the areas of bone defects compared with the other groups. Remnants of the collagen membranes were still detected in all Groups, but in Group D, the membrane seemed to have collapsed into the defect. At week 8 (Figure 18), in Groups A and B, the larger areas of newly formed bone were detected in some areas within the scaffold frameworks. However, no bone–scaffold integration was detected in either group. New bone bridging of the defects was found only in Group C. In Group D, newly formed bone was detected along the areas of the collagen membrane remnants and there was even less collagen formation within the defect areas.

## 4. Discussion

This study evaluated three major parameters that are clinically relevant to cell-based therapy: the scaffold, the stem cells and the culture medium. This is the first study to combine the PCL–BCP TDP scaffold with buccal fat ADSCs cultured in XSFM and evaluated in vitro and in animal models. For the scaffolds, the PCL and BCP raw materials used for the processing were medical grade and suitable for clinical use. When using the melt-blend processing method, the two components were physically blended without the use of any solvents that might be toxic to the cells. The BCP particles were homogenously dispersed throughout the PCL matrix, as demonstrated by the AR staining. However, the composition of 30% BCP was the maximum amount of filler that could be extruded to become filaments. In the processing, the average diameter of the PCL–BCP filaments was 1.7 ± 0.05 mm, which is in the range of the standard size of 1.74 mm of polymeric filaments used for general 3D printers. In this study, the design of the 3D printed scaffold was still based on the architecture and the interconnecting pore structure of the MSMD scaffold. The scaffolds were designed to have an interconnecting pore system and a pore size of 500 µm to allow vessel and new bone regeneration [38]. The compressive strength of the superior aspects of the scaffolds was 2.98 ± 0.01 MPa, which is comparable to that of human cancellous bone [39], and they could withstand compressive forces of up to 300 N. For the lateral aspects, the scaffolds reached 30% strain at the maximum load of 108.7 ± 4.06 N. Therefore, their mechanical strength would be adequate against wound contraction during the soft tissue healing process. The FTIR analysis indicated separate phases of BCP and PCL without chemical bonding. In addition, the FE-SEM demonstrated poor interfacial adhesion between the materials. An immiscible blend of biodegradable BCP filler and the PCL matrix created several voids inside the filaments that might accelerate degradation of the scaffolds. In animal models, our previous study demonstrated that degradation of the PCL-20% BCP MSCM scaffolds was 30.06 ± 10.48% volume loss over 90 days [3]. The BCP particles distributed throughout the filaments of the scaffolds and exposed on their surfaces would increase the bioactive charges in the calcium–phosphate crystals, confirmed by EDX analysis. Several studies indicated that calcium–phosphate crystals potentially interact with various types of stem cell attached to the materials and can regulate functions of the cells [40,41,42,43]. In this study, the PCL–BCP TDP scaffolds induced minimal inflammatory response during the first 2 weeks and new bone could regenerate into the interconnective spaces after 4 weeks. This implied that the scaffolds were biocompatible and suitable for use as an osteoconductive framework.

The adipose tissue from buccal fat pads is easily harvested with minimal tissue site morbidity, and the process is accepted by patients. In this study, the amount of fat tissue harvested from each patient was adequate for the isolation of the stem cells for all of the experiments. The plastic adherence method is cheap, practical and most commonly used for isolating stem cells from several types of tissue. After the processes of enzymatic digestion and centrifugation, the adipose tissue generates a pellet of stromal vascular fraction (SVF). SVF is a heterogeneous mixture of cells that includes ADSCs, vascular endothelial progenitors, pericytes, fibroblasts, smooth muscle cells and various blood cells (44, 45). However, only a small population of ADSCs in SVF, varying from less than 1% to over 15%, can be obtained using this method [44,45]. In the clinical applications of stem cells, a 10^5^–10^6^ cells/kg/dose is required for therapeutic levels [46,47]. To optimize cell-based therapies, small amounts of the isolated stem cells must be expanded in appropriate culture conditions to obtain sufficient cells. Therefore, the rapid expansion of the cells that can retain their multipotency and reduce exposure to culture reagents is the optimum requirement. In response to the previously mentioned risks of using FBS-containing media, the alternative is to use media containing human serum or xenogeneic serum-free media. Several studies indicated that human serum contains serum proteins, growth factors, growth hormones and essential nutrients for cell metabolism [48,49,50,51]. However, volumes of autologous serum are limited and usually inadequate for the entire culture period. In addition, serum from different donors may have different levels of the essential components. XSFM are composed of synthetic and human-derived purified substances without xenogeneic serum supplements. The composition and concentration of the substances are consistent without batch-to-batch variations. Therefore, the media are safe and suitable for clinical applications. Several studies demonstrated that MSCs from different sources, which were expanded in XSFM, potentially retained their phenotypic gene expression, proliferation and multi-differentiation during the multi-passage expansion, similar to those in the FBS-containing medium [30,31,32,33,34,35,36]. Interestingly, cells grown in XSFM had more spindle-shaped morphology, which would allow them to be grown in higher densities. Therefore, a greater number of cells can be obtained in a shorter period and can reach their confluence more rapidly. Currently, various commercial XSFM are available for expansion of stem cells. The MesenCult^TM^ medium used in this study is one of the most frequently used media for stem cell culture. Jena et al. assessed the effect of different media, including the MesenCult^TM^ medium, on the population doubling time of bone marrow MSCs [52]. The result showed that the cells in P0, which were cultured in the MesenCult^TM^ medium, had the highest proliferation when compared with the other media (*p* < 0.05). Hoang et al. assessed the efficacy of the MesenCult^TM^ medium for expansion of umbilical cord, bone marrow and adipose-derived MSCs [53]. The results demonstrated that the cells from these different sources retained their biomarker expression from the early to later passages. In addition, they differentiated into osteogenic, adipogenic and chondrogenic lineages. Shahla et al. evaluated the MesenCult^TM^ medium for in vitro expansion of ADSCs as a preliminary protocol for clinical use [30]. The cells were isolated and expanded for five passages in the Mesencult^TM^ medium and FBS-supplemented DMEM. The results demonstrated that the population doubling time of the cells cultured in the Mesencult^TM^ medium was significantly faster than those cultured in the FBS-containing medium (*p* < 0.05). In addition, the cells cultured in the Mesencult^TM^ medium had higher differentiation potential toward osteogenic and adipogenic lineages when compared with those cultured in the FBS-containing medium.

With regard to the consensus between the International Society for Cellular Therapy (ISCT) and the International Federation for Adipose Therapeutics and Science (IFATS) [54], ADSCs should be at least 80% positive to CD13, CD29, CD44, CD73, CD90 and CD105, but less than 2% positive to CD31, CD45 and CD235a. Several studies hypothesize that some subsets of adipose mesenchymal stem cells may arise from the neural crest and are pericytic in origin [55,56,57,58,59]. Therefore, CD271 and 146 are considered to be the specific markers for isolation of MSCs from adipose tissue. Some studies reported that the amount of CD271-positive cells isolated from human subcutaneous adipose tissue was 2.89 to 4.4%, corresponding with the results of our study [60,61]. Therefore, this implies that CD 271- and 146-positive cells were the subpopulation of the majority of ADSCs isolated from the buccal fat tissue. With regard to the result, the cells of both groups expressed the hematopoietic markers at less than 2% and the amount of the CD-73-positive cells of both groups was greater than 80%. However, the CD 90- and 105-positive cells were less than 80% and did not meet the IFATS criteria. This might indicate a heterogeneous population and/or the immune-stimulated status of the cells [62]. Interestingly, the percentage of CD 105 in the XSF group was only 2.94 ± 1.29%, which was significantly less than that in the FBS group. This phenomenon corresponded with the results of previous studies [63,64,65]. Brohlin et al. compared the MesenCult^TM^ medium and the minimum essential medium-alpha (α-MEM)-containing FBS for culturing ADSCs and bone marrow MSCs (64). The result indicated that the levels of CD73 and CD90 expressed from the cells in both media were not significantly different. However, the levels of CD105 in the cells, which were cultured in the MesenCult^TM^ medium, significantly reduced at passage ten. Decreased expression of CD105 at late passages was also demonstrated in the other studies when MSCs were cultured in the StemPro^TM^ MSC SFM Xeno-free medium (Thermo Fisher Scientific, Waltham, MA, USA) [63,65]. Regarding the results of our study, the growth of the cells in the XSF group was notably higher than that in the FBS group at every observation time point over 21 days. This corresponded to data from several studies, which demonstrated that MSCs cultured in the MesenCult^TM^ medium grew faster than those cultured in FBS-supplemented media [30,36,64,66]. However, Brohlin et al. reported that growth rates of bone-marrow-derived MSCs cultured in the MesenCult^TM^ medium declined after five passages due to cellular senescence during culture [64]. For cell differentiation, most of the osteogenic differentiation genes of the cells in the XSF–OS medium upregulated higher than those in the FBS–OS medium during the first 7 days. The levels of ALP, Runx-2, BSP, OPN and BMP-2 upregulated from the cells in the XSF medium were significantly higher than those of the cells in the FBS medium. It is known that Runx-2 and BMP-2 are the essential transcriptional regulators of osteogenesis, whereas ALP is induced in early calcification during osteoblast development. However, matrix mineralization of the cell–scaffold constructs was not assessed in this study because the calcium and phosphate layers of the BCP filler might confound the result of the positive staining. In vivo, the result demonstrated that the new bone volumes in the defects of Group A that were implanted with the cell–scaffold constructs and Group B that were implanted with the scaffolds alone were notably greater than those of Group D that comprised the empty defects. The newly formed bone in Groups A and B occurred at the middle portions of the defects and filled most areas in the roofs of the defects within 8 weeks. However, the amount of new bone in Group A was lower than that in Group B at all time points. This result contrasted with the findings of several previously mentioned studies. It implies that the osteoconductive property and bioactivity of the PCL–BCP TDP scaffolds were more dominant than the efficacy of the ADSCs. A possible reason might be that there was less homogeneity in the cells that were isolated using the plastic adherence method and rapidly entered the senescence phase of the ADSCs during the culture periods, and this would affect the functions of the cells in the living tissue. These factors must, therefore, be taken into account for stem cell therapy.

## 5. Conclusions

The 3D printed PCL–BCP scaffolds were biocompatible and suitable for use as the osteoconductive framework. The XSF medium was proved to support the proliferation and differentiation of ADSCs in vitro. Although the cell–scaffold constructs had no benefit in enhancing new bone formation in animal models, the scaffolds and the medium are still practical for further clinical studies and applications.

## Figures and Tables

**Figure 1 jfb-13-00093-f001:**
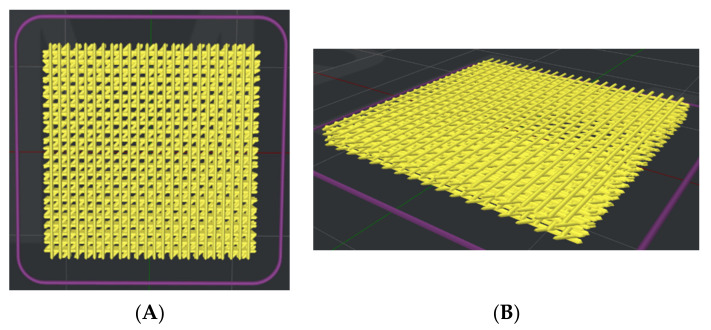
The preview architectures of the scaffold prior to the printing process; (**A**) top view and (**B**) perspective view.

**Figure 2 jfb-13-00093-f002:**
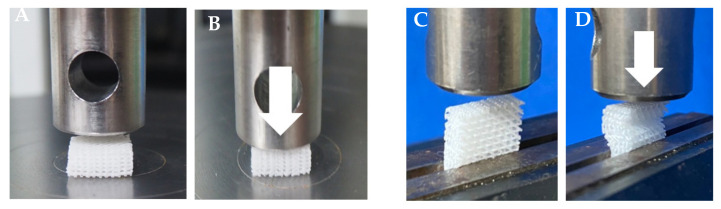
The compression forces (indicated by arrows) were applied to the superior aspect (**A**,**B**) and the lateral aspect (**C**,**D**) of the scaffolds.

**Figure 3 jfb-13-00093-f003:**
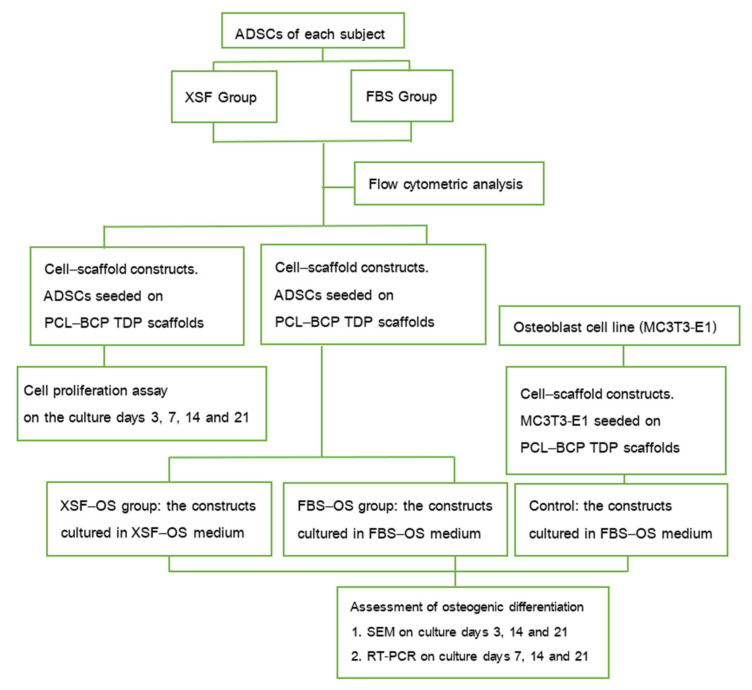
A schematic diagram showing an overview of the in vitro experiments.

**Figure 4 jfb-13-00093-f004:**
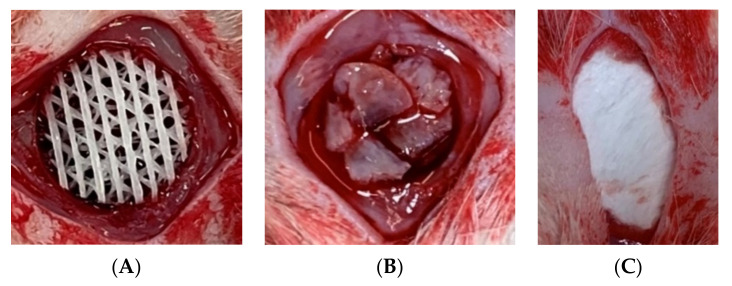
The pictures demonstrate the surgical sites of Group A (**A**) and Group C (**B**). The defects were covered with collagen membrane before suturing (**C**).

**Figure 5 jfb-13-00093-f005:**
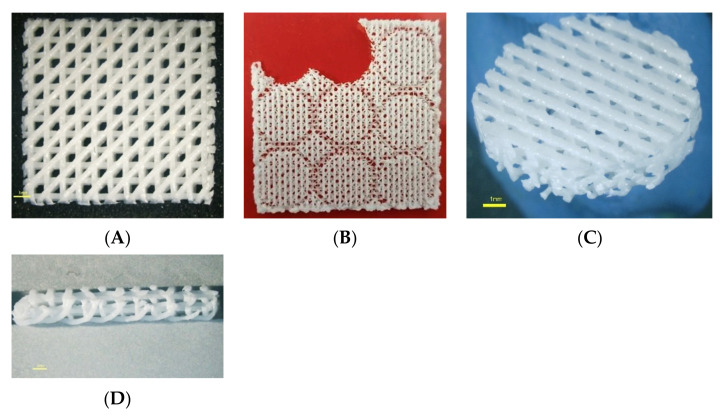
The architecture of the PCL–BCP TDP scaffold; (**A**) top view, (**B**) the scaffold was cut into round-shaped specimens for the experiments, (**C**) magnified picture of the scaffold specimen and (**D**) magnified picture of the lateral aspect of the scaffold. The scale bars represent 1 mm.

**Figure 6 jfb-13-00093-f006:**
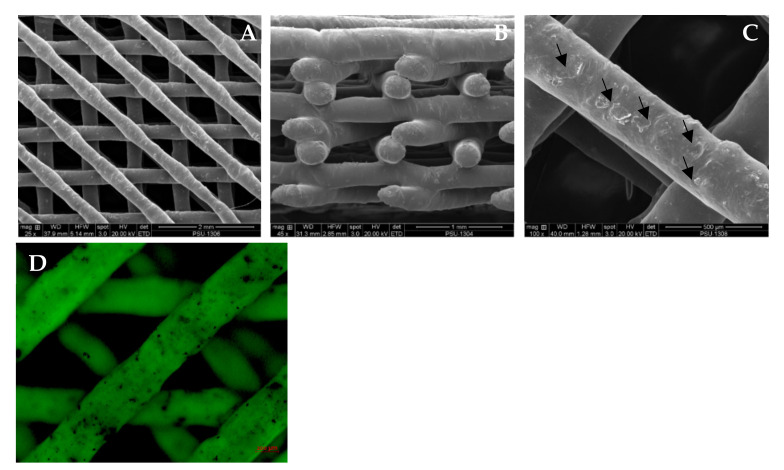
The SEM pictures demonstrate the architectures of the PCL–BCP TDP scaffolds. (**A**) top view, (**B**) lateral view and (**C**): image shows the BCP particles depositing on the surfaces of the scaffold (arrows). (**D**): The AR-stained BCP particles are seen as black spots throughout the surfaces of the scaffolds in the fluorescent microscope image.

**Figure 7 jfb-13-00093-f007:**
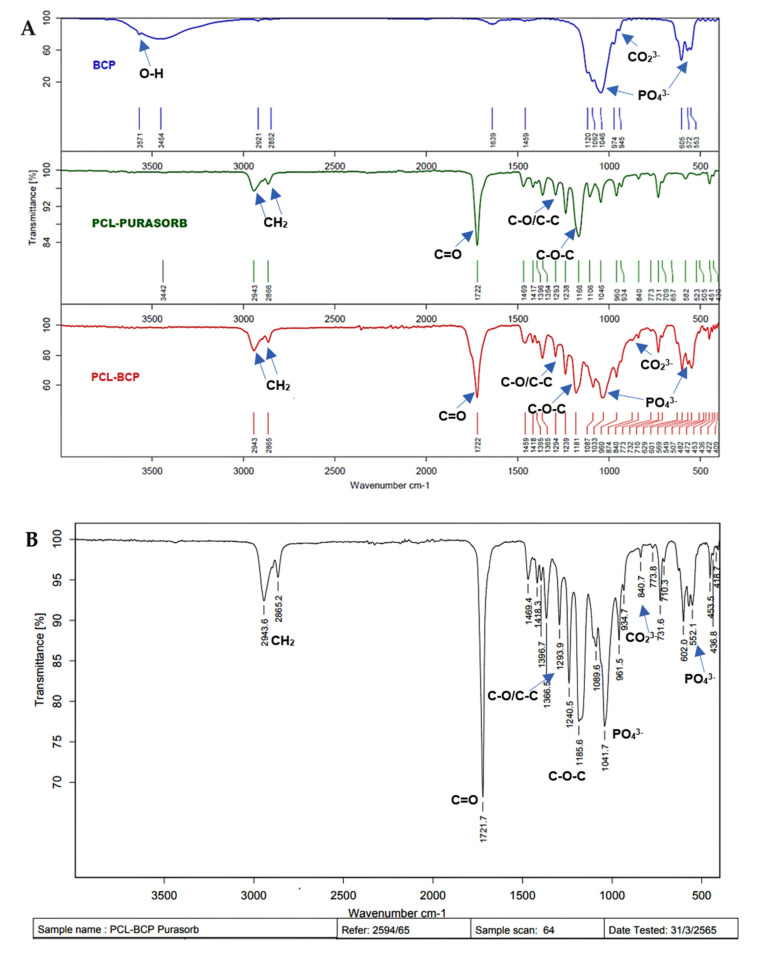
FTIR spectra for BCP, PCL and PCL-30%BCP composite. (**A**) The bands of the PCL–BCP composite showed no change when compared with the peaks of BCP and PCL raw materials. (**B**) The bands of the PCL-BCP composite were not changed when compared to those of each material.

**Figure 8 jfb-13-00093-f008:**
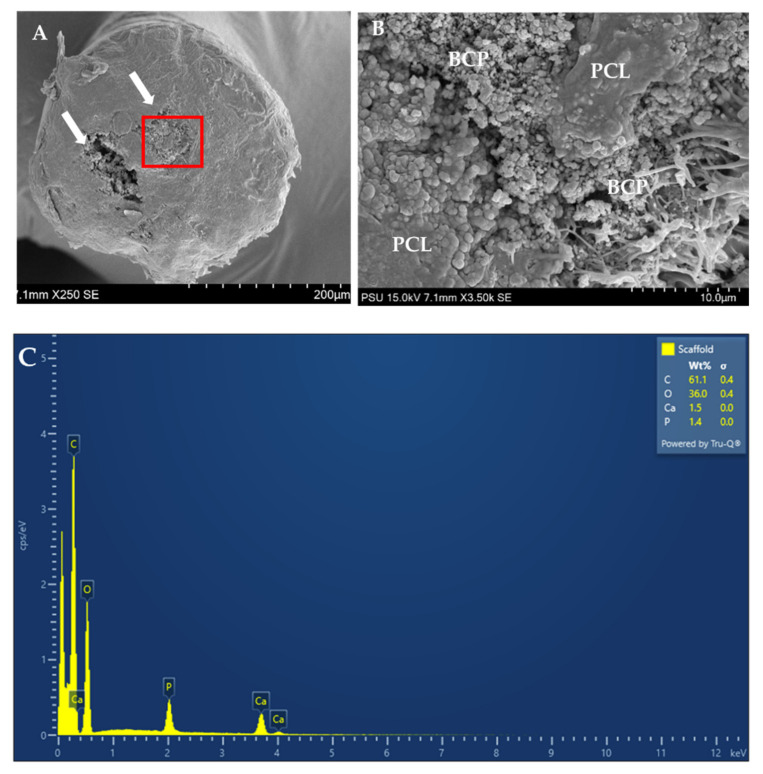
(**A**) The cross-sectional SEM image shows large voids inside the scaffold filament (indicated by arrows). (**B**) The magnified image of the box demonstrates immiscible blending of the BCP crystals and the PCL matrix in the box. (**C**) EDX analysis shows the high peaks of calcium (Ca) and phosphate (P) on the scaffold surfaces.

**Figure 9 jfb-13-00093-f009:**
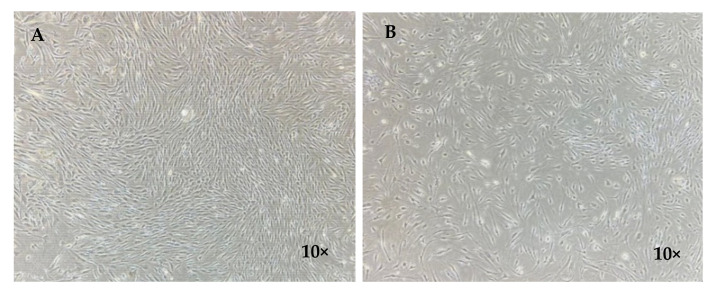
Morphologies of the adherent cells at day 21 taken via a phase-contrast microscope (DS-Fi2-U3, Nikon, Tokyo, Japan) with magnification 10×. (**A**) XSF group and (**B**) FBS group. More spindle-shaped morphologies and higher density of the cells in the XSF group were detected compared with those in the FBS group.

**Figure 10 jfb-13-00093-f010:**
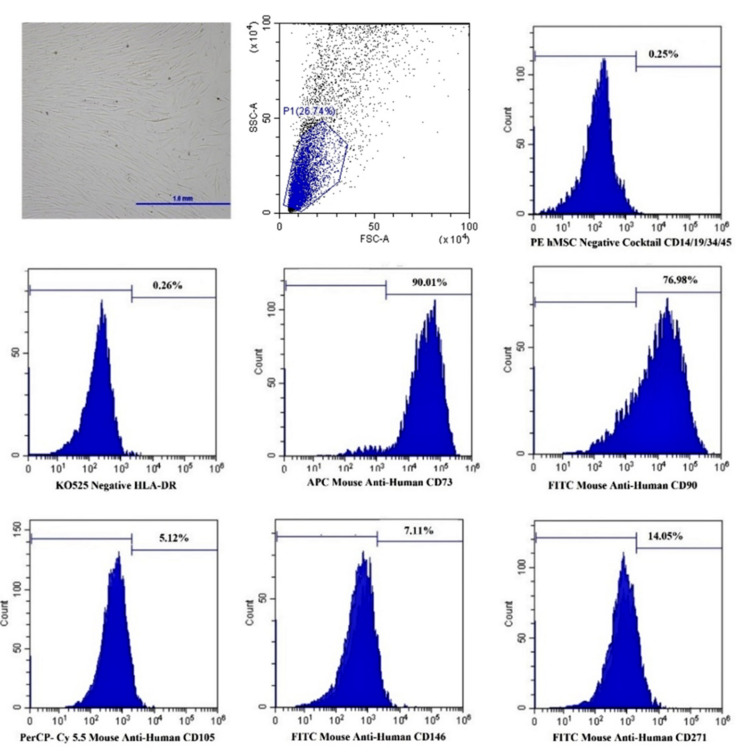
The images of flow cytometry analysis demonstrated the expression profiles of the MSC markers, hematopoietic markers, CD271 and CD146 in the XSF group.

**Figure 11 jfb-13-00093-f011:**
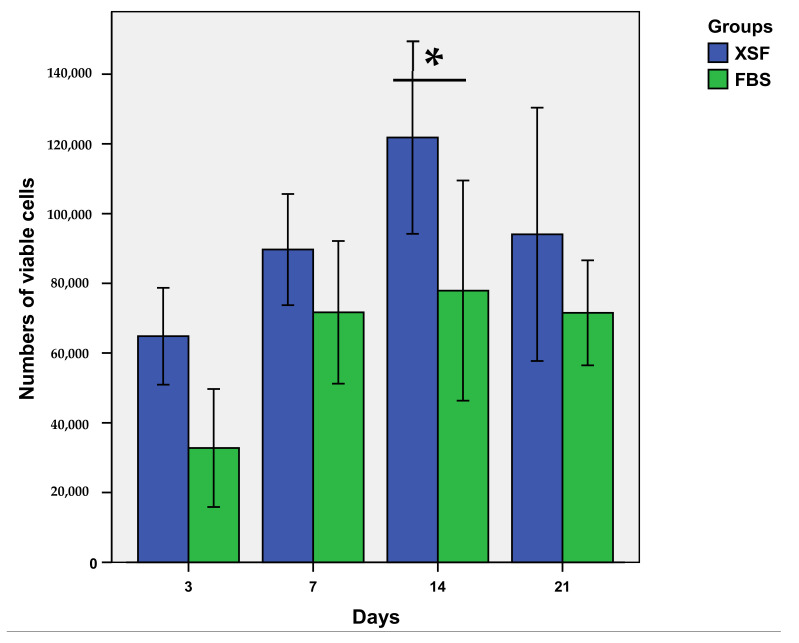
The bar graph shows the numbers of viable cells in the constructs in the XSF and FBS groups over 21 days. The growth of the cells in the XSF group was higher than that of the cells in the FBS group at all time points. The significant difference was detected at day 14 (* *p* < 0.05).

**Figure 12 jfb-13-00093-f012:**
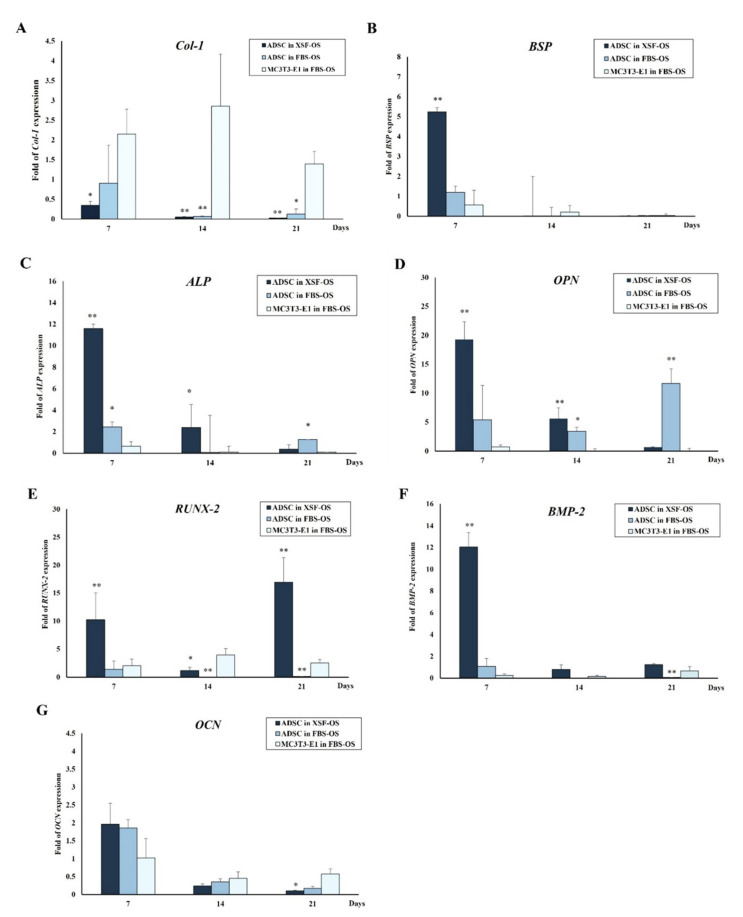
The fold change of gene expression of the cells in the constructs over 21 days; (**A**) Col-1, (**B**) BSP, (**C**) ALP, (**D**) OPN, (**E**) RUNX-2, (**F**) BMP-2 and (**G**) OCN. On day 7, the levels of the osteogenic differentiation genes in the XSF–OS group, with the exception of Col-1, were notably higher than those in the FBS–OS and control groups. The genes significantly downregulated on day 14. The significant differences were at *p* < 0.05 (*) and *p* < 0.01 (**), compared with the control group.

**Figure 13 jfb-13-00093-f013:**
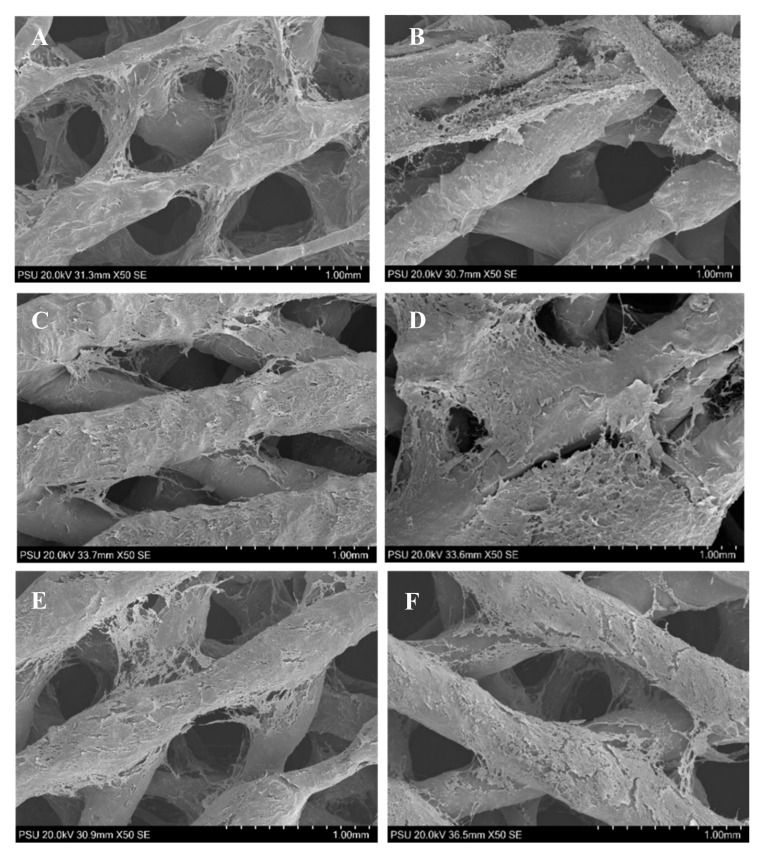
The SEM images of the cell–scaffold constructs in the XSF–OS group (**A**,**C**,**E**) and the FBS–OS group (**B**,**D**,**F**) at culture days 3 (**A**,**B**), 14 (**C**,**D**) and 21 (**E**,**F**). The cells in both groups attached and grew well on the scaffold surfaces. Dense multilayer cell sheets were observed throughout the scaffolds from day 14 and the morphologies of the cells were difficult to identify.

**Figure 14 jfb-13-00093-f014:**
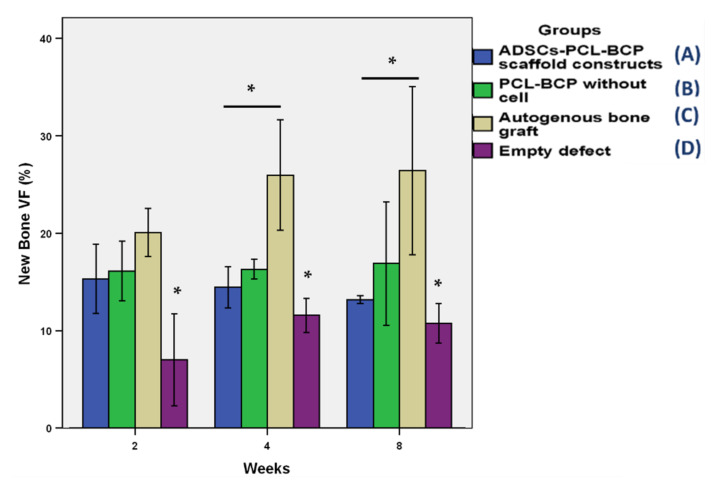
The bar graph demonstrates the new bone VFs for all groups. The new bone VFs in Group C were greater than those in the other groups, whereas those in Group D were less than the other groups over the observation period (*, *p* < 0.05 against group C). From week 4, the new bone volumes in Group C were significantly greater than those in Group A (*, *p* < 0.05). The new bone formation in Group A was less than that in Group B at all time points (*p* > 0.05).

**Figure 15 jfb-13-00093-f015:**
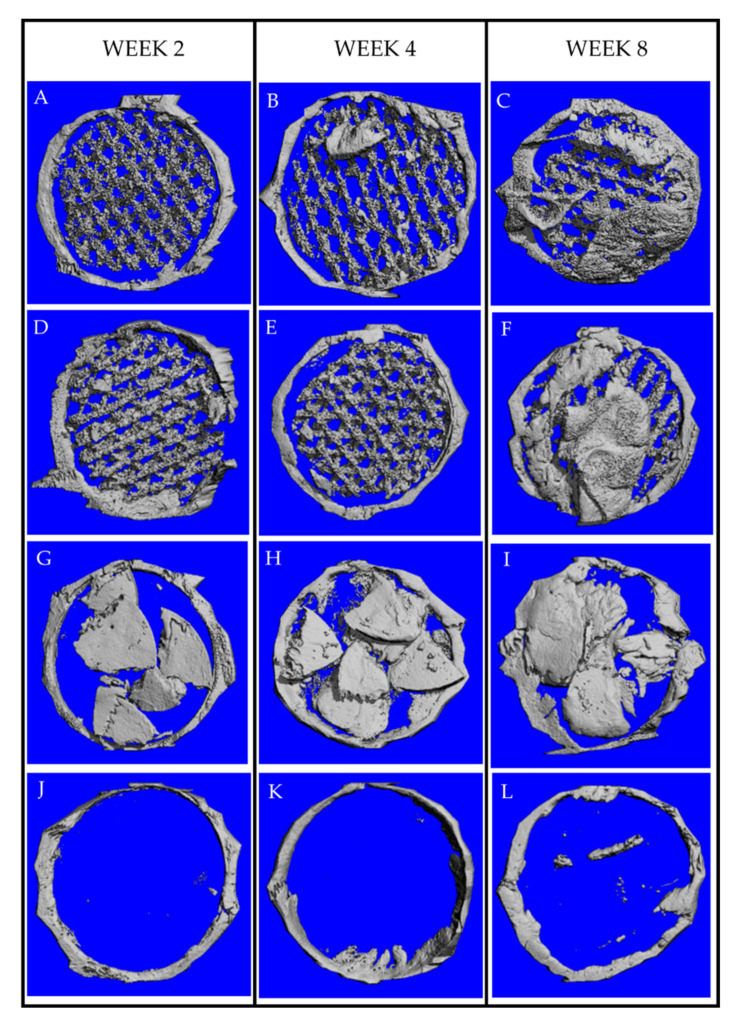
The µ-CT-constructed images demonstrate new bone formation within the defects; (**A**–**C**): Group A, (**D**–**F**): Group B, (**G**–**I**): Group C, and (**J**–**L**): Group D. The new bone formation in Groups A–C was clearly greater than in Group D. At week 8, the newly formed bone in Group A–C almost filled the entire roof of the defects, whereas in Group D, some new bone foci were detected in the middle part of the defects.

**Figure 16 jfb-13-00093-f016:**
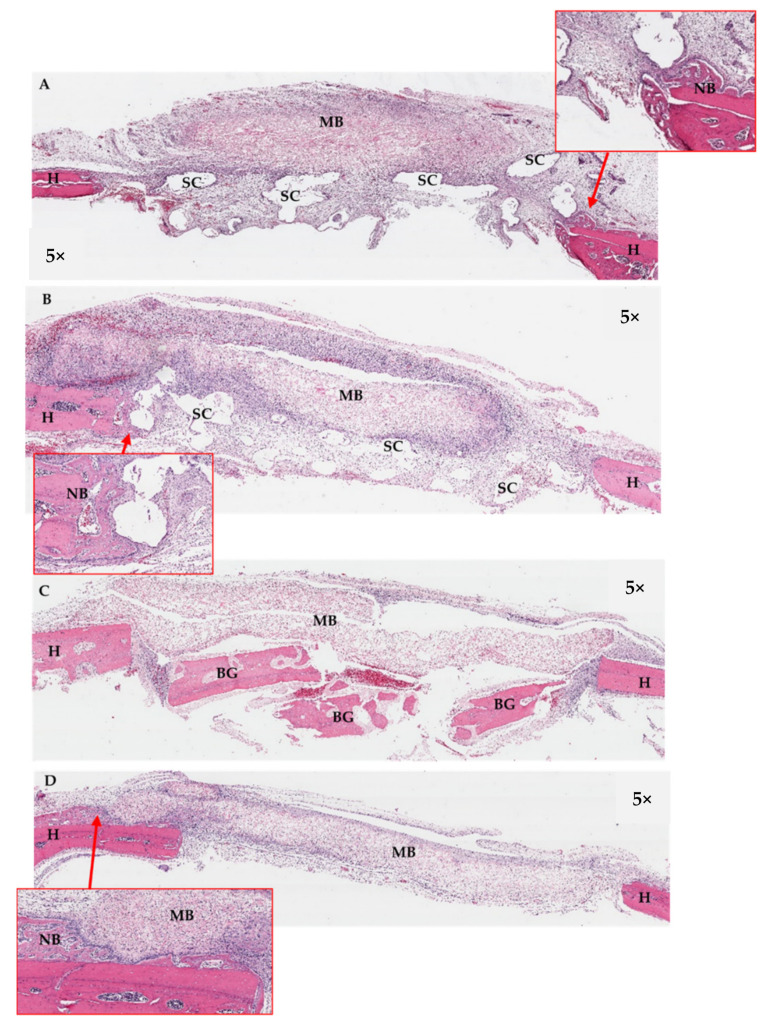
Histological features of the calvarial defects at week 2; (**A**): Group A, (**B**): Group B, (**C**): Group C, and (**D**): Group D. In Groups A and B, the scaffolds and the covering collagen membranes were surrounded by dense fibrous tissue and chronic inflammatory cells. New bone regeneration was detected extending from the periphery host bone (see boxes). In Group C, bone graft fragments were observed along the defect, which had less inflammatory cell infiltration. In Group D, newly formed bone was detected at the margins of the host bone. H = host bone, NB = new bone, SC = scaffold, MB = the collagen membranes, BG = bone graft.

**Figure 17 jfb-13-00093-f017:**
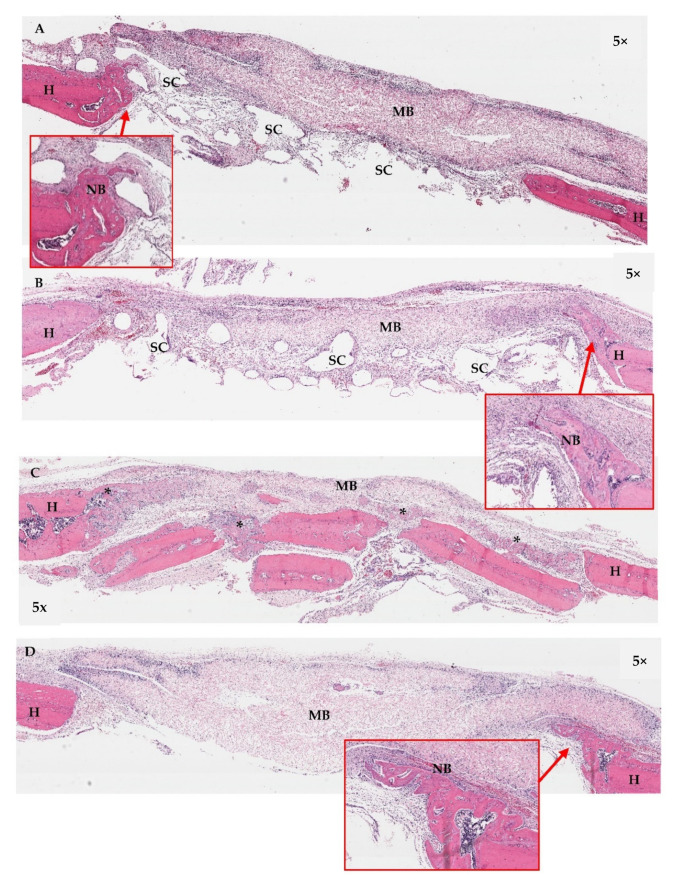
Histological features of the calvarial defects at week 4; (**A**): Group A, (**B**): Group B, (**C**): Group C, and (**D**): Group D. The areas of new bone formation within the defects in Groups A–C were larger than in week 2. In Groups A and B, the newly formed bone came from the peripheries and seemed to regenerate along the roofs of the defects (see boxes), whereas that in Group C was generally found within the middle portions of the defect (*). Remnants of the collagen membranes (MB) were still detected in all Groups. In Group D, newly formed bone regenerated from the peripheries of the defects (see box). H = host bone, NB = new bone, SC = scaffold.

**Figure 18 jfb-13-00093-f018:**
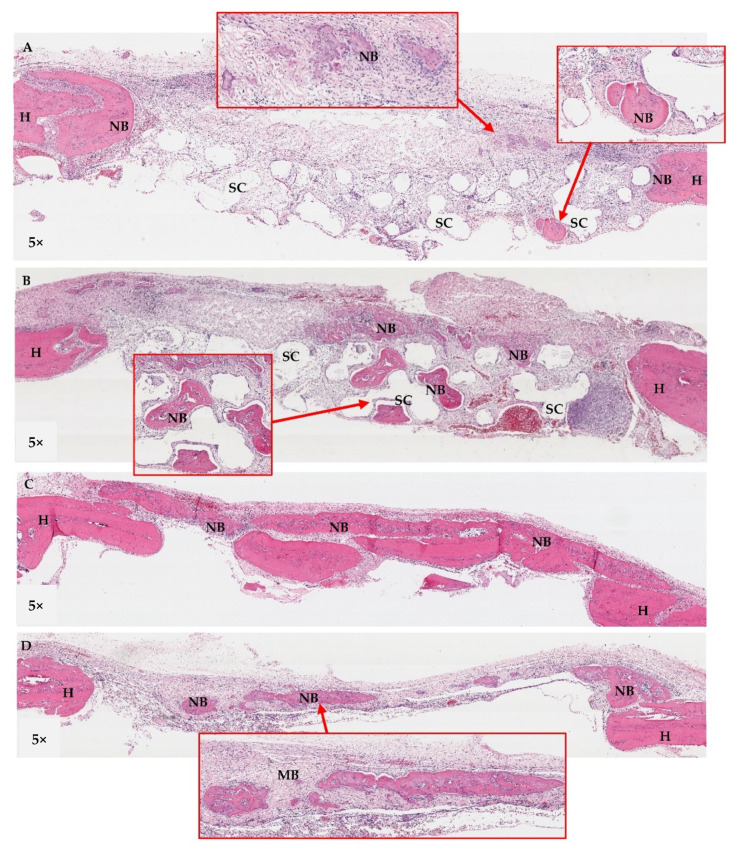
Histological sections of the calvarial defects at week 8; (**A**): Group A, (**B**): Group B, (**C**): Group C, and (**D**): Group D. In Groups A and B, the larger areas of newly formed bone were detected in some areas within the scaffold frameworks (see boxes). New bone bridging of the defects was found only in Group C. In Group D, newly formed bone was detected along the areas of the collagen membrane remnants (MB). H = host bone, NB = new bone, SC = scaffold.

**Table 1 jfb-13-00093-t001:** The oligonucleotide primer sequences of the osteogenic differentiation genes.

Genes	Primer Sequence (5′-3′)	GenBank Accession No.
Collagen type 1 (Col-1)	R: ACCAGGTTCACCGCTGTTAC	NC_000017.11
	F: GTGCTAAAGGTGCCCAATGGT	
Bone sialoprotein (BSP)	R: AGGATAAAAGTAGGCATGCTTG	NC_000004.12
	F: ATGGCCTGTGCTTTCTCAATG	
Alkaline phosphatase (ALP)	R: GCGGCAGACTTTGGTTTC	NM_001127501
	F: CCACCAGCCCGTGACAGA	
Runt-related transcription factor 2 (RUNX-2)	R: TGCTTTGGTCTTGAAATCACA	NC_10472
	F: TCTTAGAACAAATTCTGCCCTTT	
Osteocalcin (OCN)	R: CTTTGTGTCCAAGCAGGAGG	NM_00582.2
	F: CTGAAAGCCGATGTGGTCAG	
Bone morphogenetic protein-2 (BMP-2)	R: AAGAGACATGTGAGGATTAGCAGGT	NM_007553
	F: GCTTCCGCCTGTTTGTGTTTG	
Osteopontin (OPN)	R: TGTGAGGTGATGTCCTCGTCTGT	NM_00582.2
	F: ACACATATTGATGGCCGAAGGTGA	
GAPDH	R: CCACCACCCTGTTGCTGTA	NM_001289745.1
	F: GCATCCTGGGCTACACTGA	

**Table 2 jfb-13-00093-t002:** The mechanical properties of the scaffolds.

Mechanical Properties	Superior Aspect	Lateral Aspect
Compressive strength (MPa)	2.98 ± 0.01	2.18 ± 0.09
Strain at maximum load (%)	36.53 ± 1.8	30
Young’s modulus (MPa)	13.68 ± 1.01	34.75 ± 3.52
Maximum load (N)	300	108.7 ± 4.06

**Table 3 jfb-13-00093-t003:** Demographic data of the subjects.

Case	Age	Gender	Fat Volume (mL)	Operations
1	24	Male	5	BSSRO setback
2	21	Female	3	Surgical removal #18
3	23	Female	3	Surgical removal #28
4	36	Female	5	BSSRO advancement
5	21	Male	4	BSSRO setback
6	26	Female	5	BSSRO setback

**Table 4 jfb-13-00093-t004:** The average percentages of immunophenotyping markers of the cells from all subjects. The percentage of CD-105-positive cells in the FBS group was significantly higher than that of the XFS group (* *p* < 0.05).

CD Markers (%)		FBS Group	XSF Group
MSC markers	CD 73	80.12 ± 8.57%	81.26 ± 7.06%
	CD 90	66.26 ± 8.17%	67.28 ± 8.04%
	CD 105	41.58 ± 8.11% *	2.94 ± 1.29%
Hematopoietic markers	CD14, 19, 34, 45	0.16 ± 0.22%	0.25 ± 0.20%
	HLA-DR	0.42 ± 0.14%	0.28 ± 0.08%
	CD271	7.54 ± 7.10%	6.50 ± 3.45%
	CD146	4.08 ± 2.25%	4.14 ± 1.94%

## Data Availability

The data presented in this study are available on request from the corresponding author.

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
