# Peer review of "A Three-Dimensional Printed Polycaprolactone–Biphasic-Calcium-Phosphate Scaffold Combined with Adipose-Derived Stem Cells Cultured in Xenogeneic Serum-Free Media for the Treatment of Bone Defects"

_jfb, 2022, doi:10.3390/jfb13030093_

Round 1
Reviewer 1 Report
The authors studied the performance of biphasic scaffolds of polycaprolactone and calcium phosphate using in-vitro and animal models. While the manuscript is generally well executed, there are several issues that should be addressed before further consideration for publication.
1. Suggest the authors to use ISO/ASTM terminology to describe the additive manufacturing or 3D printing process used to fabricate the scaffolds.
- Khan et al. (2022), From 3D printed molds to bioprinted scaffolds: A hybrid material extrusion and vat polymerization bioprinting approach for soft matter constructs, Materials Science in Additive Manufacturing 1 (1), 7
- Serdeczny et al. (2022), Viscoelastic simulation and optimisation of the polymer flow through the hot-end during filament-based material extrusion additive manufacturing, Virtual and Physical Prototyping 17 (2), 205-219
- Ng et al. (2021), Fabrication and Characterization of 3D Bioprinted Triple-layered Human Alveolar Lung Models, International Journal of Bioprinting 7 (2), 332
2. What are the parameters used for the 3D printing, for example, layer thickness, extrusion rate etc? The parameters have effect on the properties of the scaffolds fabricated and should be clearly discussed. Any characterisation done on the scaffolds to compare them to the digital design? Are they accurate in terms of dimensions etc?
3. For the mechanical testing, any standards followed or reference for benchmarking?
4. In Table 2, some of the measurements are without standard deviations? Why is this so? Are the experiments repeatable?
Author Response
Answers for Reviewer 1
Comments and Suggestions for Authors
The authors studied the performance of biphasic scaffolds of polycaprolactone and calcium phosphate using in-vitro and animal models. While the manuscript is generally well executed, there are several issues that should be addressed before further consideration for publication.
- Suggest the authors to use ISO/ASTM terminology to describe the additive manufacturing or 3D printing process used to fabricate the scaffolds.
- Khan et al. (2022), From 3D printed molds to bioprinted scaffolds: A hybrid material extrusion and vat polymerization bioprinting approach for soft matter constructs, Materials Science in Additive Manufacturing 1 (1), 7
- Serdeczny et al. (2022), Viscoelastic simulation and optimisation of the polymer flow through the hot-end during filament-based material extrusion additive manufacturing, Virtual and Physical Prototyping 17 (2), 205-219
- Ng et al. (2021), Fabrication and Characterization of 3D Bioprinted Triple-layered Human Alveolar Lung Models, International Journal of Bioprinting 7 (2), 332
Answer: The description of the 3D printing process was reviewed in the section 2.1. The terminology of the printing parameters was revised. (Page 3, lines 115-122)
- What are the parameters used for the 3D printing, for example, layer thickness, extrusion rate etc? The parameters have effect on the properties of the scaffolds fabricated and should be clearly discussed. Any characterisation done on the scaffolds to compare them to the digital design? Are they accurate in terms of dimensions etc?
Answer: The printing parameters were described in the section 2.1. In this study, we did not compare the accuracy of the digitally designed architectures to the real architectures of the scaffolds. However, by observation, the scaffolds had been printed well using the setting parameters that mentioned in the section 2.1. Figures 5 and 6 showed that the scaffolds achieved the same architectures as the digital design. By using the scale bars of the SEM images, the diameters of the filaments are approximately 250 µm, and the pore sizes were 500 µm.
In my opinion, the parameters of the design that have most effect on the properties of the scaffolds are the pore size and the interconnecting pore system of the scaffold. In this study, the scaffolds were designed to have a pore size of 500 µm to allow vessel and new bone regeneration. The infill angle of the lay-down layer was 0◦- 45◦- 90◦ that aiming to prevent cell lost during the cell-seeding process. From the results, this design does not affect the mechanical properties of the scaffolds. Their mechanical strength is still in the acceptable range.
- For the mechanical testing, any standards followed or reference for benchmarking?
Answer: The result of the previous study1 that fabricated the pure PCL scaffolds using Fused deposition modeling process can be used for benchmarking. One of the lay-down patterns of the scaffolds was 0o/60o/120o and the compressive strength of the scaffolds was assessed. The result showed that their compression yield strength was 3.15±0.14 MPa, which is slightly higher than that of our PCL-BCP scaffolds (2.98 ± 0.01 MPa). Therefore, it implies that adding 30% of the BCP filler in the PCL-based scaffolds and the design of the scaffolds do not affect their mechanical strength.
Reference
- Zein I, et al. Fused deposition modeling of novel scaffold architectures for tissue engineering applications. Biomaterials 23 (2002) 1169–1185.
- In Table 2, some of the measurements are without standard deviations? Why is this so? Are the experiments repeatable?
Answer: The strain at maximum load of the lateral aspect and the maximum load of the superior aspect of the scaffold are the constant parameters for the compression test. As in the section 2.3, the compression force that applied to the superior aspect was limited at 300 N, whereas that applied to the lateral aspect was stopped when the strain level reached 30%.
Reviewer 2 Report
This manuscript presents an in vitro and in vivo study of 3D printed PCL–BCP TDP scaffold seeded with ADSCs, aiming bone regeneration/formation. It is well written, planned, and relevant for the audience of the journal and biomaterial community. I only have some minor comments:
- Figure 2C shows the BCP particles depositing on the scaffolds surface, and the cross-section presented in Figure 7B demonstrates immiscible blending of the BCP crystals and the PCL matrix. Please comment on this fact;
- Figures 1 and 2 should be removed from M&M section and placed on the Results;
- Please identify the peaks presented in FTIR (Figure 6);
- Please comment on Ca/P ratio, obtained from EDX analysis.
Author Response
Answers to Reviewer 2
Comments and Suggestions for Authors
This manuscript presents an in vitro and in vivo study of 3D printed PCL–BCP TDP scaffold seeded with ADSCs, aiming bone regeneration/formation. It is well written, planned, and relevant for the audience of the journal and biomaterial community. I only have some minor comments:
- Figure 2C shows the BCP particles depositing on the scaffolds surface, and the cross-section presented in Figure 7B demonstrates immiscible blending of the BCP crystals and the PCL matrix. Please comment on this fact;
Answer: The melt-blend processing without the use of solvents is very safe for cells and tissue. However, it has some points that must be concerned. Firstly, the maximum amount of the BCP filler is 30% due to increasing filament fracture and obstruction during the printing process. Secondly, the BCP particles and the PCL matrix are immiscible blend that has poor interfacial adhesion between the materials. This phenomenon created several voids inside the filaments that might reduce tensile strength of the PCL-BCP filaments. However, the result of the study showed that it did not affect the compressive strength of the PCL-BCP scaffolds. In my opinion, dispersion of the biodegradable BCP fillers and several voids inside the hydrophobic PCL matrix would accelerate degradation of the scaffolds. In the revised manuscript, the picture of Alizarin Red S-stained scaffolds was added in Figure 6 D (Page 10) for clearly demonstrating the BCP deposited on the surfaces of the scaffolds and the details of the discussion part of this answer were revised in the discussion part on page 23, lines763-768 and 778-783.
- Figures 1 and 2 should be removed from M&M section and placed on the Results;
Answer: Figures 1 and 2 were placed of the results to be Figures 5 and 6 on pages 9 and 10.
- Please identify the peaks presented in FTIR (Figure 6);
Answer: The peaks of the FTIR were identified. Figure 7 B was added to confirm that the bands of the PCL-BCP composite were not changed. (Page 11)
- Please comment on Ca/P ratio, obtained from EDX analysis.
Answer: From the result, the calcium/phosphate ratio of the PCL-30%BCP scaffolds was 1.07 (Ca = 1.5, P=1.4 Wt%). This ratio is less than that of pure hydroxyapatite (HA, Ca/P = 1.667) and pure β-tricalcium phosphate (TCP, Ca/P = 1.5)1. In our study, the EDX of the pure BCP filler (HA:TCP=30:70) was not assessed as the positive control. In addition, only 30% of the BCP filler were in the scaffold. Therefore, the Ca/P ratio of the PCL-BCP scaffolds may be lower than those of pure HA and TCP.
It is known that the Ca/P ratio plays a very important role in the mechanical properties of the biomaterials2. In my opinion, the Ca/P ratio of the BCP filler did not affect the mechanical strength of the scaffold due to the small amount of the filler that composed in the PCL matrix.
References:
- U Tariq, et al. Calcium to phosphate ratio measurements in calcium phosphates using LIBS. Journal of Physics: Conf. Series 1027 (2018) 012015.
- Raynaud S, et al. Determination of Calcium/Phosphorus Atomic Ratio of Calcium Phosphate Apatites Using X‐ray Diffractometry. Journal of the American Ceramic Society 2001, 84(2) 359–366.
Reviewer 3 Report
Comments to authors
The in vitro findings of the present study describes that the adipose derived stem cells (ADSCs) cultured in xenogeneic serum free media (XSFM) are an excellent source of cells for new bone formation. Also, the in-house fabricated PCL-BCP scaffolds are biocompatible and suitable for use as an osteoconductive framework. However, there is no significant enhancement of new bone formation was observed in cell-scaffold construct. The following are few minor revisions required before publication.
Minor revisions
1. From the results, it is evident that the cell-scaffold construct did not significantly enhance new bone formation compared to scaffold alone groups. Hence, it is recommended that the author should change the title of the manuscript “A three-dimensional printed polycaprolactone–biphasic calcium phosphate scaffold combined with adipose-derived stem cells cultured in xenogeneic serum-free media for the treatment of bone defects”.
2. Include appropriate references for the following:
· For line 46, “Its biocompatibility has been widely demonstrated in several in vivo and clinical studies”.
· For line 54, “BCP consists of the stable phase of HA and the more soluble phase of β- tricalcium phosphate (β-TCP)”.
3. Keep the formatting same for sub-sections: 2.2, 2.3 as in 2.1(italicized without bold).
4. The scale bar in Figure 1 is not visible. Please resize the same. Scale bar missing in Figure 8.
5. Clarity of Figure 11 should be improved. Also keep the font size same for all the description.
6. In section 2.8.2, the authors described that 36 male Wistar rats were used for the experiment. In line 307, they mentioned n=4/group/time point. In total there are 4 groups (A, B, C and D) and bone formation evaluated at time points (2, 4 and 8 weeks). This counts to a total of 48 Wistar rats. Please re-gcheck and make it clear, in total how many animals were used for the experiment?
7. In Figure 14, mark 2, 4 and 8 weeks for ease of identification. Also arrange the image panels neatly.
8. Image size of all the panels in Figure 15 should be same. Add scale bar.
9. The manuscript lack information on funding support. It is mandatory to include the same as per the journal’s guideline.
Author Response
Answers to Reviewer 3
Comments and Suggestions for Authors
Comments to authors
The in vitro findings of the present study describes that the adipose derived stem cells (ADSCs) cultured in xenogeneic serum free media (XSFM) are an excellent source of cells for new bone formation. Also, the in-house fabricated PCL-BCP scaffolds are biocompatible and suitable for use as an osteoconductive framework. However, there is no significant enhancement of new bone formation was observed in cell-scaffold construct. The following are few minor revisions required before publication.
Minor revisions
- From the results, it is evident that the cell-scaffold construct did not significantly enhance new bone formation compared to scaffold alone groups. Hence, it is recommended that the author should change the title of the manuscript “A three-dimensional printed polycaprolactone–biphasic calcium phosphate scaffold combined with adipose-derived stem cells cultured in xenogeneic serum-free media for the treatment of bone defects”.
Answer: Initially, regarding the results of our previous studies and the other literature, the combination of the ADSCs and the PCL-BCP scaffolds was hypothesized that it could enhance new bone formation both in vitro and in vivo better than using the scaffolds alone. Using the xenogeneic serum-free media for culturing the cell-scaffold constructs was planned for the further clinical trial. In this study, the results of the in vitro experiments supported that hypothesis. However, the result in the animal model was not met the hypothesis due to the factors that I mentioned in the discussion part (Page 25, lines 886-890). Therefore, in my opinion, the name of the title is appropriate for describing the aims of the study.
- Include appropriate references for the following: · For line 46, “Its biocompatibility has been widely demonstrated in several in vivo and clinical studies”. · For line 54, “BCP consists of the stable phase of HA and the more soluble phase of β- tricalcium phosphate (β-TCP)”.
Answer: I added the references 7-12 for those sentences (Page 2, line 47 and 55).
- Keep the formatting same for sub-sections: 2.2, 2.3 as in 2.1(italicized without bold).
Answer: Done
- The scale bar in Figure 1 is not visible. Please resize the same. Scale bar missing in Figure 8.
Answer: The new scale bars in the figures 5A, C and D (Page 9) were added at the same sizes of the original images and added “The scale bars represent 1 mm.” in the caption. In figure 9, unfortunately, the scale bars were not included at the time of taking the images. However, the detail of magnification “Morphologies of the adherent cells at day 21 taken via a phase-contrast microscope (DS-Fi2-U3, Nikon, Japan) with magnification 10X.” was added (Page 13, line 457-458). (Regarding the comment of another reviewer, Figures 1 and 2 were placed on the results to be Figure 5 and 6. Figure 8 was changed to be Figure 9)
- Clarity of Figure 11 should be improved. Also keep the font size same for all the description.
Answer: The font size of the caption was corrected to be the same size. All graphs of the figures 12 were enlarged and increased quality. (Figure 11 was changed to be Figure 12 on page 16)
- In section 2.8.2, the authors described that 36 male Wistar rats were used for the experiment. In line 307, they mentioned n=4/group/time point. In total there are 4 groups (A, B, C and D) and bone formation evaluated at time points (2, 4 and 8 weeks). This counts to a total of 48 Wistar rats. Please re-check and make it clear, in total how many animals were used for the experiment?
Answer: The number of animals in section 2.8.2 was changed to be 48 (Page 8, line 279).
- In Figure 14, mark 2, 4 and 8 weeks for ease of identification. Also arrange the image panels neatly.
Answer: The figure 15 was revised; the images were arranged into the column of each time point and re-sized. (Figure 14 was changed to be Figure 15 on page 19)
- Image size of all the panels in Figure 15 should be same. Add scale bar.
Answer: The images of figures 16-18 were re-sized. The scale bars were not included at the time of taking the images, therefore, I added magnification of 5x on each image. (Figures 15-17 were changed to be Figure 16-18 on pages 20-22)
- The manuscript lack information on funding support. It is mandatory to include the same as per the journal’s guideline.
Answer: The information on funding support was added in the funding section (Page 25, lines 909-910).
Round 2
Reviewer 1 Report
NA